# The Fluorescent Quenching Mechanism of N and S Co-Doped Graphene Quantum Dots with Fe^3+^ and Hg^2+^ Ions and Their Application as a Novel Fluorescent Sensor

**DOI:** 10.3390/nano9050738

**Published:** 2019-05-13

**Authors:** Yue Yang, Tong Zou, Zhezhe Wang, Xinxin Xing, Sijia Peng, Rongjun Zhao, Xu Zhang, Yude Wang

**Affiliations:** 1Department of Physics, Yunnan University, Kunming 650091, China; Yangyue_018@163.com (Y.Y.); wzz0307@yeah.net (Z.W.); xingxinxin@126.com (X.X.); rjzhao0504@hotmail.com (R.Z.); 2School of Materials Science and Engineering, Yunnan University, Kunming 650091, China; zoutong626@163.com (T.Z.); sjpeng18@163.com (S.P.); zhangxubrian@163.com (X.Z.); 3Key Lab of Quantum Information of Yunnan Province, Yunnan University, Kunming 650091, China

**Keywords:** N and S co-doped graphene quantum dots, fluorescence quenching, DFT, non-radiative electron transfer, inner filter effect, fluorescence sensor

## Abstract

The fluorescence intensity of N, S co-doped graphene quantum dots (N, S-GQDs) can be quenched by Fe^3+^ and Hg^2+^. Density functional theory (DFT) simulation and experimental studies indicate that the fluorescence quenching mechanisms for Fe^3+^ and Hg^2+^ detection are mainly attributed to the inner filter effect (IFE) and dynamic quenching process, respectively. The electronegativity difference between C and doped atoms (N, S) in favor to introduce negative charge sites on the surface of N, S-GQDs leads to charge redistribution. Those negative charge sites facilitate the adsorption of cations on the N, S-GQDs’ surface. Atomic population analysis results show that some charge transfer from Fe^3+^ and Hg^2+^ to N, S-GQDs, which relate to the fluorescent quenching of N, S-GQDs. In addition, negative adsorption energy indicates the adsorption of Hg^2+^ and Fe^2+^ is energetically favorable, which also contributes to the adsorption of quencher ions. Blue fluorescent N, S-GQDs were synthesized by a facile one-pot hydrothermal treatment. Fluorescent lifetime and UV-vis measurements further validate the fluorescent quenching mechanism is related to the electron transfer dynamic quenching and IFE quenching. The as-synthesized N, S-GQDs were applied as a fluorescent probe for Fe^3+^ and Hg^2+^ detection. Results indicate that N, S-GQDs have good sensitivity and selectivity on Fe^3+^ and Hg^2+^ with a detection limit as low as 2.88 and 0.27 nM, respectively.

## 1. Introduction

Graphene quantum dots (GQDs) have drawn research interest by reasons of their excellent photoluminescence, low toxicity, and good biocompatibility [1]. They have been applied in various fields, such as solar cells, light-emitting diodes, bioimaging, and fluorescent sensors [2]. Some reports have shown that the fluorescence intensity of GQDs can be quenched by Ag^+^ and Hg^2+^ ions [3,4,5]. According to the interaction between quantum dots and a quencher, the main quenching mechanism can be divided into Förster resonance energy transfer (FRET), the inner filter effect (IFE), and the dynamic and static quenching process [6,7]. The quenching process requires sufficient contact between quantum dots and a quencher. Diffusive encounters collision contact can be attributed to the dynamic quenching process, while new complex forms can be assigned to the static quenching process [8]. In a typical static quenching process, the interaction between a quencher and quantum dots can form a non-fluorescent complex. While in the dynamic quenching process, energy or charge transfer between a quencher and the excited states of quantum dots can lead to fluorescence quenching. For IFE quenching, the fluorescence quenching requires the absorption spectra of quantum dots overlapping with the excitation or emission spectra of quantum dots in the presence of a quencher. However, for FRET quenching, it requires the emission spectra of fluorescent quantum dots overlapping with the absorption spectrum of a quencher [7]. Some reports have indicated that the good affinity between metal ions and heteroatom doped GQDs is conductive to the electron transfer between quantum dots and a quencher. However, the contribution of doped atoms for fluorescence quenching is less detailed [9,10]. On the other hand, it is difficult for experimental research to give the atomic interaction details between a quencher and GQDs. Density functional theory (DFT) simulation has been applied to study the atomic and electronic properties of materials [11,12]. It can give more detail about the atomic information of materials. Hence, to understand the contribution of doped atoms for fluorescent quenching, the DFT simulations can be applied to investigate the relationship between a quencher and N, S-GQDs.

Based on the fluorescent quenching properties of GQDs, it can be served as a novel fluorescence probe for metal ion detection [5]. Iron is one of the essential metal elements for many physiological and pathological processes, including oxygen transport, enzyme catalysis, and hemopoiesis [1,13]. Although the Fe^3+^ ion plays an important role in the human body, overload and deficiency of ferric ions in the body can cause health problems like cancer, Parkinson’s disease, and dysfunction of organs [13,14,15]. After years of investigation of the relationship between Fe^3+^ and certain diseases, people have understood the essential role of iron in the human body [16,17,18]. Thus, it is critical to detect traces of Fe^3+^. Unlike ferric ions, mercury ions are well-known as one of the extremely toxic substances. Mercury ions can bioaccumulate in the human body from the environment through the food chain causing serious health problems, such as renal failure and nervous system damage [19,20]. The emissions of mercury are mainly due to the heavy use of fossil fuels [21,22]. Some studies have shown that even 5 ppb of mercury ions can lead to serious damage of organs [20,21]. Given this fact, it is significant to track the amount of ferric and mercuric ions. Fluorescent sensor have been applied to detect heavy metal ions and it is characterized by simple preparation, rapid response, and high sensitivity [9,23].

Therefore, in this study, DFT simulations were used to investigate the interaction between a quencher (Fe^3+^ and Hg^2+^) and N, S-GQDs. The relationship between the fluorescent quenching mechanism and doped heteroatoms were carried out by DFT simulations and experimental studies. According to our DFT and experimental studies, the quenching mechanism can be assigned to the dynamic quenching by Hg^2+^ and the IFE quenching by Fe^3+^. Due to the electronegativity difference of doped N and S atoms, the charge distribution of surrounding C atoms can be changed. The charge redistribution of N, S-GQDs is conducive to the adsorption of cation, resulting in the effective electron transfer between N, S-GQDs and Fe^3+^ or Hg^2+^. The adsorption of Fe^3+^ or Hg^2+^ on the surface of N, S-GQDs can lower the system total energy, which also can facilitate the adsorption of cation. N, S-GQDs were synthesized by the one-pot hydrothermal method to further validate the fluorescent quenching mechanism. The results show that N, S-GQDs have excellent sensitivity and selectivity to Fe^3+^ and Hg^2+^ ions. The good affinity between the quencher (Fe^3+^ and Hg^2+^) and the surface functional groups of N, S-GQDs has a contribution to the special recognition toward Fe^3+^ and Hg^2+^, respectively. Significantly, the as-synthesized N, S-GQDs fluorescence sensor presented a good detection limit as low as 2.88 and 0.27 nM for Fe^3+^ and Hg^2+^ detection, respectively.

## 2. Materials and Methods

### 2.1. Computational Details

Density functional theory was applied to investigate the interaction of Fe^3+^ or Hg^2+^ with N, S-GQDs, respectively. The theoretical simulations were based on CASTEP code with generalized gradient approximation (GGA) and Perdew–Burke–Enzerh (PBE) of exchange-correlation functional [23,24,25,26,27]. Core electrons were modeled with the fully-relativistic ultra-soft pseudo-potentials [28]. The energy cut-off of the graphene lattice was set to 400 eV. The number of *k* points for the Brillouin zone integration was chosen according to a Monkhorst–Pack grid of 0.03 Å^−1^ with a convergence criterion of 1 × 10^−6^ eV [29]. The structures were geometrically optimized using the Broyden–Fletcher–Goldfarb­–Shanno (BFGS) method before lattice dynamical property calculation [30]. Their constituent atoms were allowed to relax until the interatomic forces became below the value of 0.03 eVÅ^−1^. The total energy convergence criterion was set to be 10^−5^ eV.

### 2.2. Materials

The chemical reagents citric acid, thioacetamide, sodium hydroxide, and phosphoric acid were provided by Aladdin Chemistry Co. Ltd. (Shanghai, China). The disodium hydrogen phosphate, sodium dihydrogen phosphate, and metal salts were purchased from Tianjin Zhiyuan Chemical Reagent Co. Ltd. (Tianjin, China). All the reagents are analytical grade and used as received.

### 2.3. Synthesis of N-GQDs

The synthesis of N, S-GQDs was through a hydrothermal process using critic acid and thioacetamide as the initial carbon, nitrogen, and sulfur source, respectively. Briefly, 0.5 g critic acid and 0.0267 g thioacetamide were dissolved into 60 mL purified water by ultrasonic treatment for 5 min. Then the mixture was transferred into 100 mL Teflon-lined autoclave and heated at 180 °C for 10 h. After being cooled down to room temperature, the suspension was centrifuged at 10,000 rpms for 15 min. The obtained N, S-GQDs solution was collected after being further purified through dialysis (cutoff molecular weight: 300 Da) for 10 h. 

### 2.4. Characterization

The morphology of as-synthesized N, S-GQDs was recorded on a JEOL-JEM 2100 transmission electron microscope (TEM) and Seiko SPA-400 SPM atomic force microscope (AFM). The optical properties of N, S-GQDs were carried out by UV-vis-1800 spectrophotometer (Jinghua Instrument, Shanghai, China). The Fourier transformed infrared spectra (FTIR) were recorded by an Avatar-360 spectrometer. K-Alpha + X-ray electron spectrometer was used to record the X-ray photoelectron spectroscopy (XPS) of N, S-GQDs. The photoluminescence spectra were conducted on a Horiba Fluorolog-3 fluorescence spectrophotometer. Fluorescence lifetime decays were acquired on a Quantaurus-Tau fluorescence lifetime spectrometer (Hamamatsu, Japan).

### 2.5. Detection of Metal Ions

The detection of Fe^3+^ or Hg^2+^ was performed by measuring the fluorescence spectra in the presence and absence of metal ions. In a typical analysis process, 200 µL N, S-GQDs (0.08 mg/mL) were dispersed into 1 mL PBS (phosphate buffered saline) buffer (0.1 M, pH 7), followed by the addition of a certain amount of Fe^3+^ or Hg^2+^, respectively. The schematic diagram of detection process and device are shown in Appendix A. Then the solution was diluted to 5 mL with PBS buffer and incubated for 10 min at room temperature. The detection of Fe^3+^ or Hg^2+^ was assessed by the fluorescence quenching ratio (I/I_0_) with various metal ion concentrations, where I and I_0_ were corresponding to the fluorescence intensity in the presence and absence of metal ions, respectively. The selectivity of N, S-GQDs was investigated by adding various certain concentrations of interfering ions (50 µM).

### 2.6. Detection of Fe^3+^ and Hg^2+^ in Real Samples

To evaluate the potential application of N, S-GQDs in real water samples detection, bottled drinking water (Nongfu Spring Co. Ltd., Hangzhou, China) were chosen as real samples. The Hg^2+^ ions were diluted to 50, 100, and 300 nM by drinking water, while the Fe^3+^ ions were diluted to 700 nM, 1 μM, and 3 μM, respectively. Then, the appropriate volume of cation solutions were skipped into the fluorescence solution, and diluted to 5 mL with PBS buffer.

## 3. Results and Discussion

### 3.1. Fluorescent Quenching Mechanisms of N, S-GQDs with Fe^3+^ or Hg^2+^

DFT simulations were carried out to investigate the quenching mechanism of N, S-GQDs on Fe^3+^ or Hg^2+^. The adsorption energy (*E*_ad_) and Mulliken populations were calculated to investigate the interaction between N, S-GQDs and a quencher. For the simulations, 18 atoms’ hexagonal configurations were applied, as shown in Figure 1. Atomic population (Mulliken) analysis indicates that carbon atoms in pure graphene are neutral, as shown in Table 1. As set out in Table 1, the S atoms have a positive charge, while the negative charges are located on N and O atoms. This can be ascribed to the electronegativity difference of N, S, and O atoms. N and O atoms have larger electronegativity and stronger charge-accepting ability than S. The positively charged S atom breaks the electro-neutral situation of surrounding C atoms, and that is conductive to the generation of charge favorable sites for the interaction of cations and N, S-GQDs [31]. The atomic populations of Fe^3+^ @N, S-GQDs and Hg^2+^ @N, S-GQDs are listed in Appendix A. The adsorption energy can be defined as the energy difference between N, S-GQDs and the adsorption system. The *E*_ad_ of Fe^3+^ and Hg^2+^ were calculated as −2.43 and −3.27 eV, respectively, which indicate the adsorption of Hg^2+^ and Fe^2+^ are energetically favorable. According to the DFT calculations, the adsorption of Fe^3+^ or Hg^2+^ change the charge distribution on N, S-GQDs as shown in Appendix A. Atomic population calculations reveal that the amount of charge transfer between N, S-GQDs and adsorbed Fe^3+^ is 0.66 e^−^ and the charge transfer for Hg^2+^ is 0.13 e^−^, respectively, implying the interaction existence between N, S-GQDs and a quencher (Fe^3+^ or Hg^2+^). Based on the DFT simulations, the quenching mechanism of N, S-GQDs by Fe^3+^ and Hg^2+^ might be attributed to the dynamic quenching.

In order to further investigate the fluorescent quenching mechanism of N, S-GQDs by Fe^3+^ or Hg^2+^, N, S-GQDs were synthesized by a hydrothermal method. In the following, the fluorescence lifetime and UV-vis spectra were measured to determine the fluorescence quenching mechanism of N, S-GQDs, as shown in Figure 2. As shown in Figure 2, the average fluorescence lifetime of N, S-GQDs without metal ions (Fe^3+^ or Hg^2+^) is 8.15 ns. While the average fluorescence lifetime of N, S-GQDs in the presence of Fe^3+^ and Hg^2+^ can be determined as 8.02 and 7.72 ns, respectively. The average fluorescence lifetime of N, S-GQDs decay after the addition of Hg^2+^, indicating the fluorescence quenching is probably attributed to the non-radiative electron transfer between N, S-GQDs and a quencher [6,32]. However, the average fluorescence lifetime in the presence of Fe^3+^ only changes slightly, which indicates the quenching process is probably related to other quenching mechanisms [33,34,35]. Figure 2d depicts the UV-vis spectra and fluorescence spectra of N, S-GQDs in the absence and presence of Fe^3+^ or Hg^2+^, respectively. The absorption threshold and peak position change cannot be observed, which indicates that no stable metal complexes form [36]. It is noticeable that the absorption spectra of N, S-GQDs with Fe^3+^ have overlaps with the emission and excitation spectra of N, S-GQDs, which is related to the characteristic of IFE quenching [7,33]. The fluorescence lifetime of N, S-GQDs changed in the presence of Hg^2+^ but no changes in the absorption spectra of N, S-GQDs can be observed. Therefore, based on the DFT and experimental analyses, the fluorescence quenching mechanism of N, S-GQDs by Fe^3+^ is mainly attributed to the IFE, while the fluorescence quenching by Hg^2+^ is related to the dynamic quenching process, as shown in Figure 3.

### 3.2. Characterization of N, S-GQDS

After the quenching mechanism analysis of N, S-GQDs, the as-synthesized N, S-GQDs were applied to detect Fe^3+^ and Hg^2+^, respectively. The morphological properties of as-synthesized N, S-GQDs were carried out by TEM and AFM images as shown in Figure 4. The high-resolution transmission electron microscopy (HRTEM) image, as shown in the inset of Figure 4a, displays the crystal lattice of N, S-GQDs with a lattice spacing distance of 2.23 Å, which corresponds to the (1120) plane of graphite [5]. The TEM image shows that the as-synthesized N, S-GQDs are well-dispersed and have a narrow sized distribution ranging from 1.5 to 4 nm with an average size of 2.5 nm, as shown in Figure 4b. AFM images reveal the thickness of N, S-GQDs, as shown in Figure 4c. The topographic height of 1.0–2.5 nm can be seen in Figure 4d with an average height of 1.5 nm. The thickness of single layered graphene is 0.34 nm, which suggests that most N, S-GQDs are single or bilayered. Therefore, it can be concluded that the as-synthesized quantum dots are graphene quantum dots.

XPS measurements were performed to characterize the elemental composition properties of as-synthesized N, S-GQDs, as shown in Figure 5. Figure 5a shows the survey spectra of N, S-GQDs. Four dominant peaks at 168.8, 285.2, 401.7, and 533.4 eV can be identified as S 2p, C 1s, N1s, and O1s, respectively. The C, S, O, and N configuration in N, S-GQDs were investigated by the deconvolution of S 2p, C 1s, N1s, and O1s peaks, as shown in Figure 5b–e. As shown in Figure 5b, the high-resolution spectrum of C 1S can be deconvoluted into four peaks at 283.9, 284.8, 285.7, and 288.4 eV, which can be assigned to C–S–C, C–C, C–O/C–H, and O–C=O, respectively [37,38]. The covalent bond of C–S–C at 283.9 eV confirms the presence of S doping in N, S-GQDs. Figure 5c represents the high-resolution spectrum of N 1s. The binding energy peaks located at 399.4 and 401.2 eV are the contributions of pyrrolic N (C–N–C) and O=C–N, respectively [39,40]. The S 2p spectrum, as shown in Figure 5d, can be deconvoluted into four distinct peaks at 163.5, 164.6, 168.1, and 169.3 eV. The peaks at 163.5 and 164.6 eV are related to the covalent bond of thiophene–S C–S–C 2p 3/2 and C–S–C 2p 1/2. The latter peaks agree with S=O (168.1 eV) and sulfone bridges C-SO_2_-C (169.3 eV) [41,42]. The appearance of pyrrolic N and C–S–C bond peaks indicates that S and N atoms have been successfully doped into GQDs. The spectrum of O 1s, as shown in Figure 5e, can be fitted with three peaks at 531.8, 532.4, and 533.7 eV, which are related to the contribution of O–H (531.8 eV), C=O/COOH (532.4 eV), and C–O (533.7 eV) bonds, respectively [43].

Figure 6 shows the FTIR spectrum of as-synthesized N, S-GQDs. The bands appearing around 3417 cm^−1^ are related to the stretching vibration of O–H or N–H bonds. The broad absorption band at 2528 cm^−1^ is assigned to the vibration of S–H. The absorption peaks at 1709 and 1403 cm^−1^, which are ascribed to the contribution of carboxyl groups (C=O) and the amide linkage (O=C–NH) bending vibration, indicate that there are abundant oxygen-contained groups on the surface of as-synthesized N, S-GQDs [40,44]. It is worthy to notice that the absorption bands at 1195 cm^−1^ are assigned to the vibration of C–S or C–N bonds, indicating that S and N atoms have been doped into the graphene configuration. Peaks at 1573 and 1673 cm^−1^ are corresponding to C=N and C=C stretching.

### 3.3. Optical Analyses of N, S-GQDS

UV-vis and fluorescence spectrum were applied to investigate the optical properties of N, S-GQDs, as shown in Figure 7. As shown in Figure 7a (blue curve), two clear characteristic absorption peaks at 230 and 350 nm can be observed, respectively. The absorption peak at 230 nm is contributed from the π–π* transition [40,45]. Typical n–π* transition absorption bands of C=O and C=N are plotted at 350 nm [5]. The fluorescence excitation wavelength of N, S-GQDs is 350 nm, as given by the photoluminescence excitation spectrum of Figure 7a (black curve), indicating that the n–π* transition is the domain excitation mode in the photoluminescence progress of N, S-GQDs [46]. The maximum wavelength of photoluminescence is 438 nm (red curve), which further proves N, S-GQDs aqueous solution emits strong blue fluorescence under UV lamp irradiation. It can be seen in Figure 7a that the fluorescence intensities of N, S-GQDs are quenched by Hg^2+^ and Fe^3+^ ions, respectively. Furthermore, the photoluminescence wavelength reveals nearly excitation-dependence properties, as shown in Figure 7b. Excitation dependent properties of N, S-GQDs are the result of surface defect and particle size variation. The emission wavelength changes slightly and is attributed to the narrow particle distribution of as-synthesized N, S-GQDs. The stabilities of as-synthesized N, S-GQDs were evaluated by recording the fluorescence intensity of N, S-GQDs for 60 days, as shown in Appendix A. The results indicate that N, S-GQDs have good fluorescent stability. Besides that, heteroatom N and S doping into graphene quantum dots’ lattice also contributes the excitation-dependence properties of N, S-GQDs [37].

### 3.4. Selectivity of N, S-GQDs for Fe^3+^ and Hg^2+^ Detection

The selectivity of N, S-GQDs were evaluated by measuring the fluorescence quenching ratio as a result of the addition of various cations, including Ag^+^, K^+^, Ba^2+^, Mg^2+^, Ca^2+^, Al^3+^, Co^2+^, Cd^2+^, Zn^2+^, Cu^2+^, Fe^2+^, Na^+^, Ni^2+^, and Pb^2+^. The fluorescence quenching ratios were recorded in the presence of 50 µM various cations, as shown in Figure 8a. Of all the cations tested, only Hg^2+^ and Fe^3+^ led to efficient quenching of fluorescence intensity. Moreover, the interfering tests of N, S-GQDs, as shown in Figure 8b, were investigated through recording the fluorescence quenching ratio on Fe^3+^ or Hg^2+^ with the addition of various interfering cations (50 µM). As displayed in Figure 8c, the presence of interfering ions does not have an obvious impact on Fe^3+^ or Hg^2+^ detection. When Fe^3+^ or Hg^2+^ coexist in the detection sample, the interfering of another cation can be eliminated by adding ascorbic acid (AA) and cysteine (Cys), respectively.

Some reports have shown that the good affinity between quencher cations and functional groups on the surface of quantum dots relates to the good selectivity of GQDs’ fluorescence probe [47,48,49]. XPS and FTIR analysis show that there are many carboxyl, hydroxyl, amino, and hydrosulfonyl groups on the surface of N, S-GQDs. Fe^3+^ and Hg^2+^ have good affinity with them. Hence, N, S-GQDs have better selectivity to Fe^3+^ or Hg^2+^ than other cations. Bond populations and electron density difference also were calculated to investigate the interaction between N, S-GQDs and a quencher (Fe^3+^ or Hg^2+^). As listed in Appendix A, the bond population of Fe-S was calculated as 0.17, indicating Fe^3+^ can covalently bond with S. The electron density difference of Fe^3+^@N, S-GQDs and Hg^2+^@N, S-GQDs are plotted in Figure 9. The red areas represent the electron enrichment and the blue areas indicate electron withdrawal. The electron density difference distributed between Fe and S atoms, as shown in Figure 9a, shows the characteristic of covalent bond distribution, which agrees with bond population analysis. According to the electron density difference and bond population of Hg^2+^@N, S-GQDs, as shown in Figure 9c,d, there are no obvious covalent or chemical bonds formed between Hg and doped atoms (N, S). The atomic population of doped S shows that a doped S atom has a positive charge population. This situation could reduce the ability for S atoms forming chemical bonds with interfering cations. Meanwhile, energetically favorable adsorption and the charge favorable sites on the surface of N, S-GQDs also avail the electron transfer between Fe^3+^, Hg^2+^, and N, S-GQDs [9]. Therefore, N, S-GQDs have better selectivity to Fe^3+^ and Hg^2+^ than other interfering cations.

### 3.5. The Fluorescence Properties of N, S-GQDs under Acidity and Alkalinity Situations

The fluorescence stability and fluorescence quenching ratio were tested by measuring fluorescence intensity change on various pH values from 2.0 to 12.0, as shown in Figure 10. The fluorescence intensities of N, S-GQDs remain stable in alkaline conditions, while there is a decrease of fluorescence intensity in the pH range from 2.0 to 7.0. The decrease of fluorescence intensity of N, S-GQDs in acidic solution can be attributed to the protonation of amino and carboxylic groups in N, S-GQDs. As the carboxyl and amino groups on the surface of N, S-GQDs tend to bond with protons, resulting in the redistribution of surface electrons in acidic conditions. The fluorescence intensity decrease is assigned to the aggregation of N, S-GQDs after the protonation process of amino and carboxylic groups take place [50]. However, the carboxyl groups on the surface of N, S-GQDs are deprotonated in alkaline medium and form a negative charge shell, which makes the fluorescence properties of N, S-GQDs retain a stable performance [51]. As shown in Figure 10, the fluorescence quenching performance of N, S-GQDs show pH-dependent behavior in the presence of 1 μM Hg^2+^ and 1μM Fe^3+^, respectively. Carboxyl groups on the surface of N, S-GQDs are well protonated in acidic medium, which can weaken the affinity between metal ions (Hg^2+^ or Fe^3+^) and carboxyl groups, resulting in less fluorescence quenching. Hence, the fluorescence quenching performance of as-synthesized N, S-GQDs has pH-dependence behavior. Considering the fluorescence intensity and best fluorescence quenching of N, S-GQDs, the following photoluminescence spectra were measured under pH 7.

### 3.6. Detection of Fe^3+^ and Hg^2+^ Using N, S-GQDs as a Sensor

The fluorescence quenching performance after the addition of various concentrations of Fe^3+^ is plotted in Figure 11. As shown in Figure 11a, the fluorescence intensities of N, S-GQDs are quenched by Fe^3+^. The fluorescence quenching ratios are performed to determine the sensitivity of as-synthesized N, S-GQDs for Fe^3+^ detection, as shown in Figure 11b–d. Figure 11b shows the fluorescence quenching ratio as a function of Fe^3+^ concentration in the range 0–100 μM. Two individual linear relationships between the fluorescence quenching ratio and the concentration of Fe^3+^ can be observed in the dynamic ranges 1–90 nM and 0.1–30 μM, as shown in Figure 11c,d, respectively. As shown in Figure 11c, the linear regression equation (I and I_0_ refer to the fluorescence intensity of N, S-GQDs at 438 nm in the presence and absence of Fe^3+^, respectively) is I/I_0_ = 0.98951–0.0005194 [Fe^3+^]. The corresponding regression coefficient (R^2^) is 0.985. A good detection limit (LOD) of 2.88 nM can be obtained by the standard curve of the fluorescence quenching ratio and Fe^3+^ concentration (1–90 nM). The detection limit is defined by a signal-to-noise ratio of 3. In addition, the LOD can be calculated as 55.49 nM in the dynamic range 0.1–30 μM. According to the Standards for Drinking Water Quality of China National Standers, the threshold limit of Fe^3+^ ion content in water is 0.3 mg/L (5.37 μM) [52]. Therefore, the dynamic range and detection limit of as-synthesized N, S-GQDs have comparable application potential for Fe^3+^ detection in drinking water.

Besides that, the as-synthesized N, S-GQDs were applied to determine Hg^2+^, as shown in Figure 12. As described in Figure 12a, the as-synthesized N, S-GQDs exhibit Hg^2+^ concentration-dependent fluorescence properties. The fluorescence intensity of N, S-GQDs can be quenched after the addition of various amounts of Hg^2+^. There is a good linear relationship (I/I_0_ = 0.99383−0.00181 [Hg^2+^], R^2^ = 0.991) between the fluorescence quenching ratio and Hg^2+^ concentration in the range 1–30 nM, as shown in Figure 12c. The detection limit can be calculated as 0.27 nM, which satisfies the requirements of the Chinese National Standards (0.001 mgL^−1^, 4.98 nM) and the United States Environmental Protection Agency (0.002 mg L^−1^, 9.96 nM) for the safety demands of Hg^2+^ in drinking water [52,53]. In addition, another linear relationship can be observed in the range 100–1000 nM with the detection limit of 36.85 nM, which also meets the requirements of the China National Standards for the upper safety limit of total mercury content in industrial effluent (0.05 mg L^−1^, 249 nM) [52]. The doped atom N and S not only has contributed to the fluorescence quenching process of N, S-GQDs, but also makes the as-synthesized N, S-GQDs have a comparable detection performance on Fe^3+^ or Hg^2+^ to other fluorescence sensing systems. The detection performances of N, S-GQDs for Fe^3+^ or Hg^2+^ are compared with various reported fluorescence sensors, as listed in Table 2 and Table 3.

### 3.7. Real Sample Detection Analyses

To evaluate the applicability of an as-synthesized N, S-GQDs fluorescence probe, the performance of N, S-GQDs in real drinking water samples was investigated. Appendix A displays the fluorescence intensity change spectra of N, S-GQDs with different concentrations of Fe^3+^ (70 nM, 1 μM, and 3 μM) and Hg^2+^ (50, 100, and 300 nM), respectively. The recoveries of Fe^3+^ or Hg^2+^ detection are given in Appendix A. Recoveries of drinking water sample detection are close to 100%. The obtained results imply that the as-synthesized N, S-GQDs have potential as a fluorescent probe for Fe^3+^ or Hg^2+^ detection in drinking water

## 4. Conclusions

In this work, DFT and experimental studies indicate that non-radiative electron transfer between Hg^2+^ cations and N, S-GQDs cause dynamic fluorescence quenching. The good sensitivity of N, S-GQDs for Fe^3+^ is mainly attributed to the IFE quenching. Doped atoms can produce charge favorable sites, which can interact with Fe^3+^ or Hg^2+^, leading to the fluorescence quenching of N, S-GQDs. *E*_ad_ calculations reveal that the adsorption of Hg^2+^ or Fe^3+^ are energetically favorable, which also contribute to the non-fluorescence electron transfer. The as-synthesized N, S-GQDs were applied to detect Fe^3+^ and Hg^2+^ in aqueous solution. The results show that as-synthesized N, S-GQDs have good selectivity and sensitivity to Fe^3+^ and Hg^2+^, respectively. The analysis of the fluorescence quenching ratio displays a good linear relationship in both Fe^3+^ and Hg^2+^ detection. For Fe^3+^ detection, a good detection limit of 55.49 nM can be found in the dynamic response range of 0.1 μM to 30 μM, which is lower than the recommended content of Fe^3+^ in drinking water by China National Standers. Furthermore, a good detection limit of 0.27 nM with a wide dynamic range (1–30 nM) was obtained for Hg^2+^ detection, implying that this method has potential for nanomolar level detection of Hg^2+^ in aqueous solution. Therefore, the facile, inexpensive, and sensitive N, S-GQDs present a promising candidate as a fluorescence probe for fluorescence detection.

## Figures and Tables

**Figure 1 nanomaterials-09-00738-f001:**
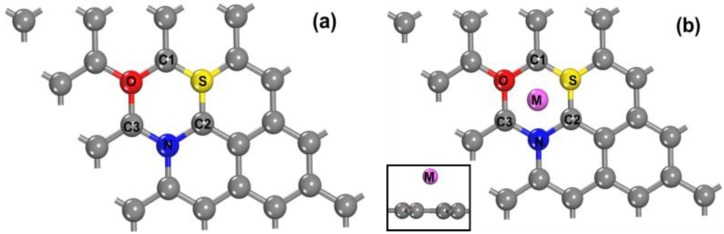
The top view of density functional theory (DFT) simulation modes (**a**) N, S co-doped graphene quantum dots (N, S-GQDs), (**b**) M cations (M = Fe^3+^, Hg^2+^) adsorb on the surface of N, S-GQDs.

**Figure 2 nanomaterials-09-00738-f002:**
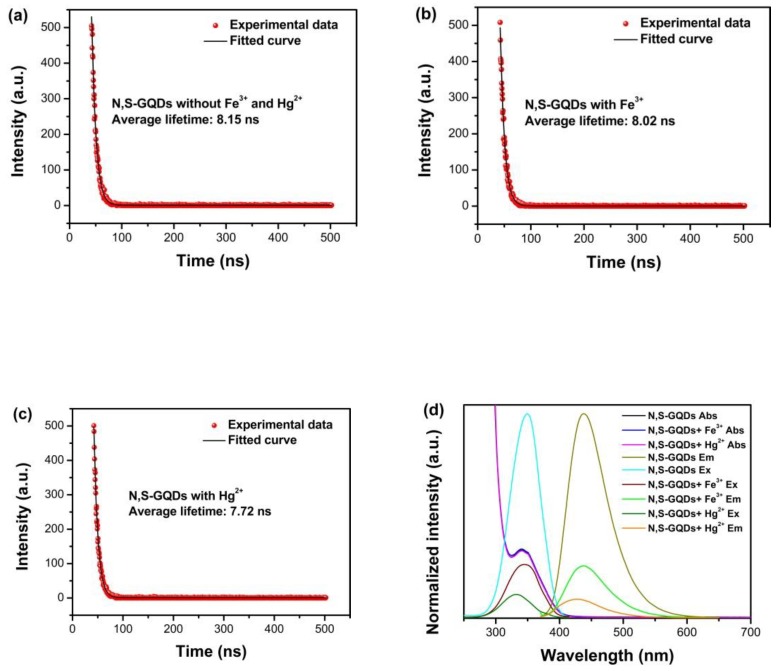
Photoluminescence decay spectra of N, S-GQDs (**a**) without cations and the addition of 10 μM Fe^3+^ (**b**) and 10 μM Hg^2+^ (**c**), respectively. (**d**) UV-vis absorption spectra (10 μM Fe^3+^ and Hg^2+^) and fluorescence spectra of N, S-GQDs in the presence and absence of 20 μM Fe^3+^ and 10 μM Hg^2+^.

**Figure 3 nanomaterials-09-00738-f003:**
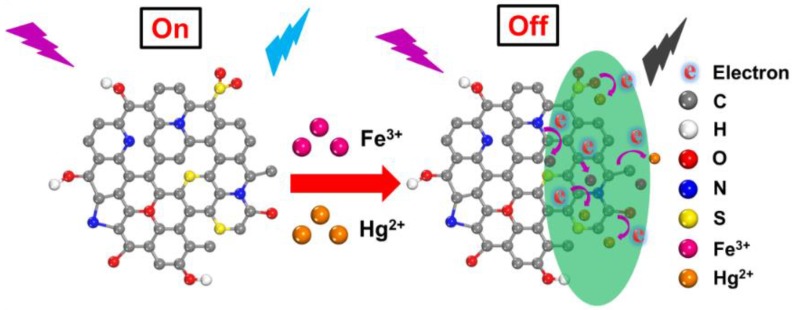
Schematic diagram of N, S-GQDs as a fluorescence probe for the detection of Fe^3+^ or Hg^2+^ ions.

**Figure 4 nanomaterials-09-00738-f004:**
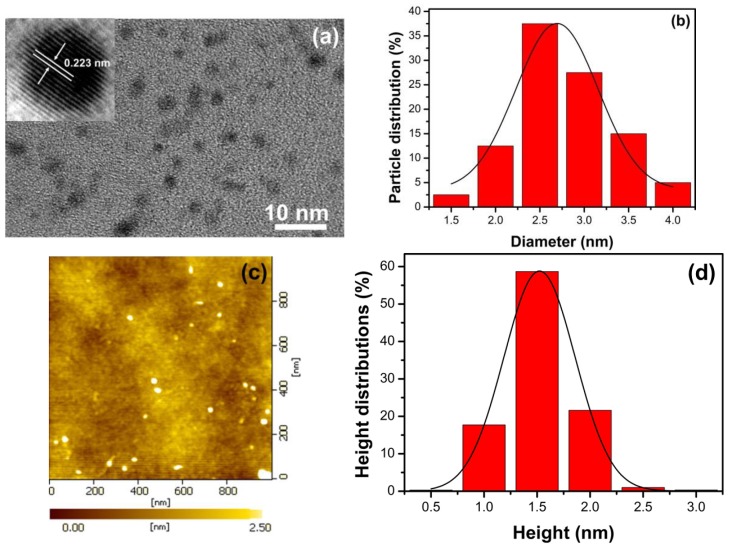
(**a**) TEM image of N, S-GQDs and HRTEM image are inserted as insets; (**b**) the corresponding particle size distribution (n = 40); (**c**,**d**) atomic force microscope (AFM) image and particle height distribution of as-synthesized N, S-GQDs.

**Figure 5 nanomaterials-09-00738-f005:**
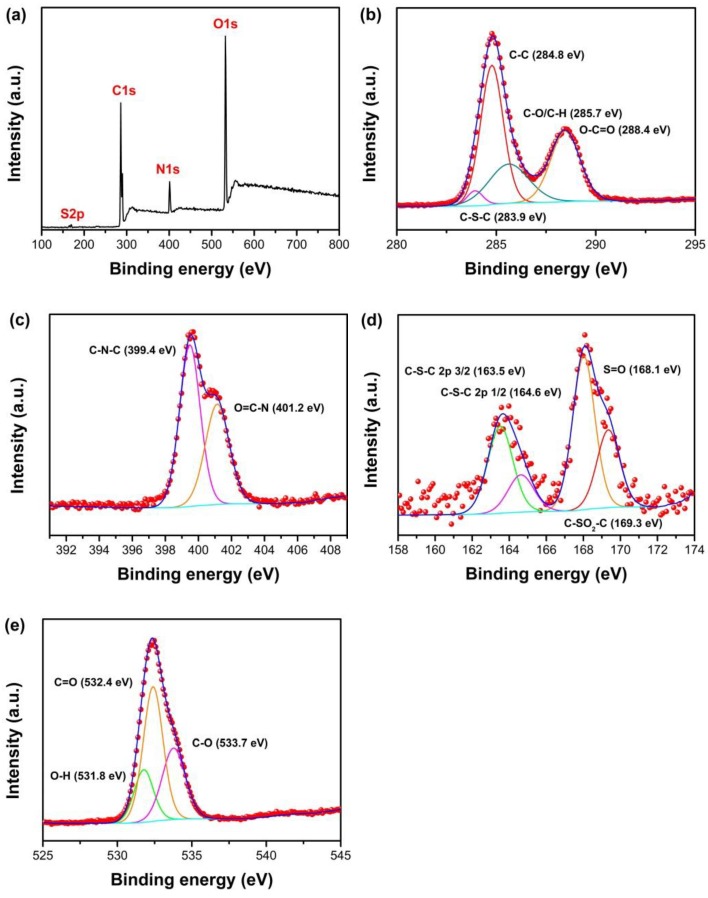
(**a**) The X-ray photoelectron spectroscopy (XPS) survey spectrum of N, S-GQDs, (**b**–**e**) high resolution spectra of C 1s, N 1s, S 2p, and O 1s of as-synthesized N, S-GQDs, respectively.

**Figure 6 nanomaterials-09-00738-f006:**
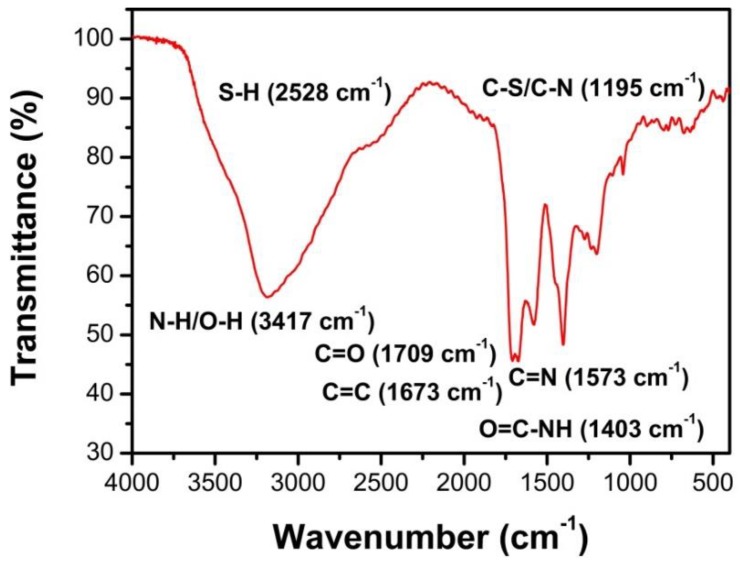
FTIR spectra of as-synthesized N, S-GQDs.

**Figure 7 nanomaterials-09-00738-f007:**
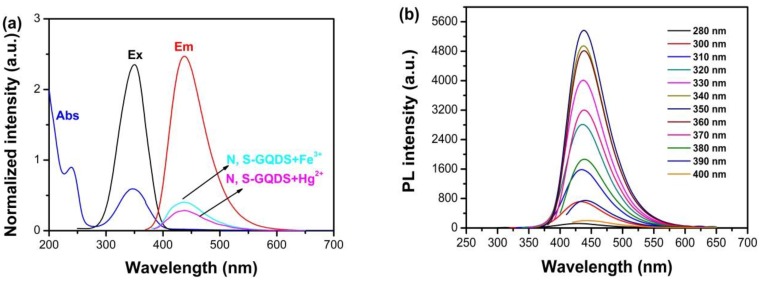
(**a**) UV-vis absorption, fluorescence excitation, and fluorescence emission spectra of N, S-GQDs; (**b**) the excitation dependent photoluminescence (PL) emission spectra of N, S-GQDs.

**Figure 8 nanomaterials-09-00738-f008:**
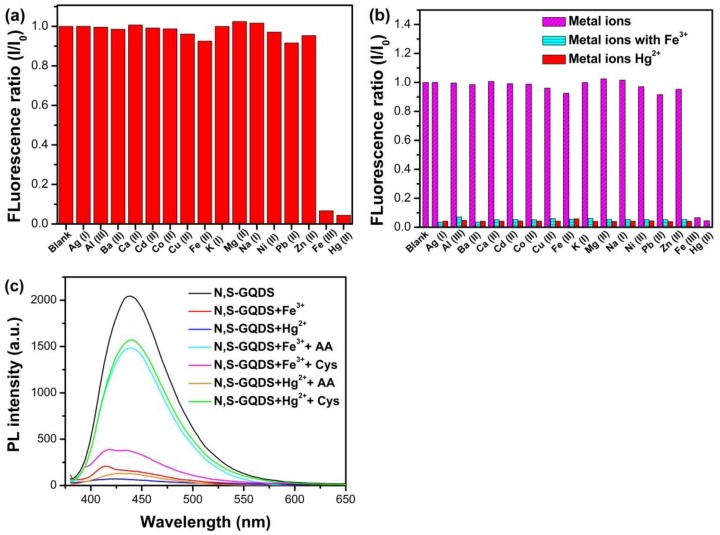
(**a**) The fluorescence quenching ratio of as-synthesized N, S-GQDs toward various cations; (**b**) the interfering test of N, S-GQDs; (**c**) the fluorescence spectra of N, S-GQDs before and after the addition of 50 μM Fe^3+^, 50 μM Hg^2+^, 100 μM AA, and 50 μM Cys, respectively.

**Figure 9 nanomaterials-09-00738-f009:**
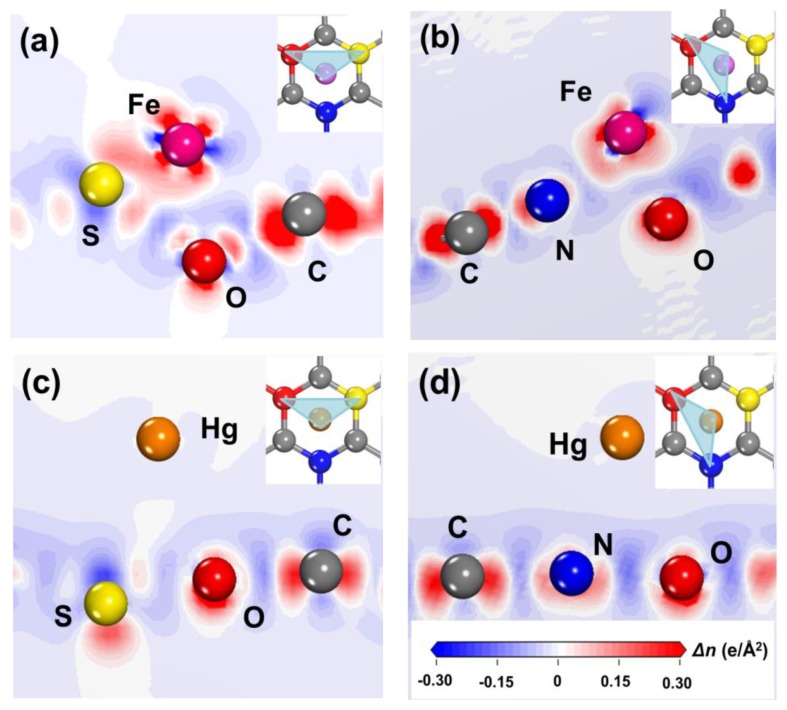
The electron density difference of N, S-GQDs after the adsorption of Fe^3+^ (**a**,**b**) and Hg^2+^ (**c**,**d**), respectively.

**Figure 10 nanomaterials-09-00738-f010:**
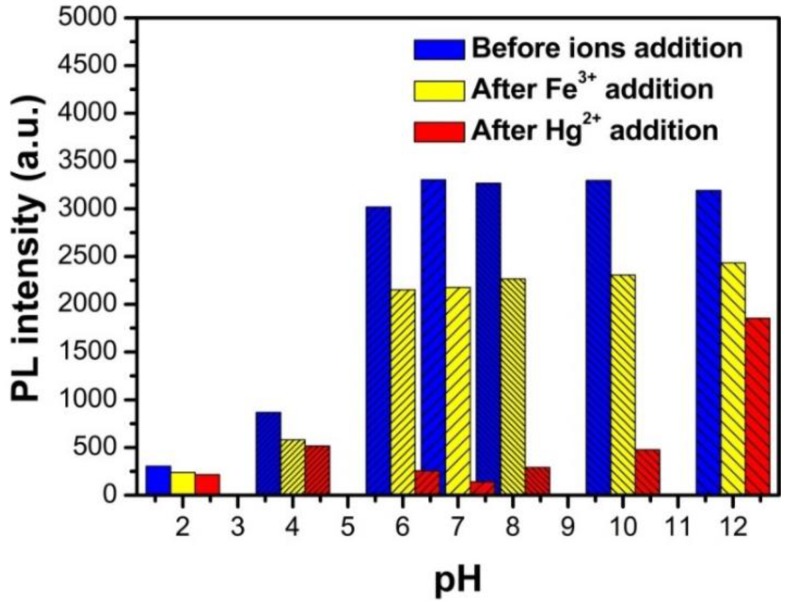
The dependence of fluorescence intensity on different pH of N, S-GQDs before and after the addition of 1 μM Hg^2+^ or 1 μM Fe^3+^, respectively.

**Figure 11 nanomaterials-09-00738-f011:**
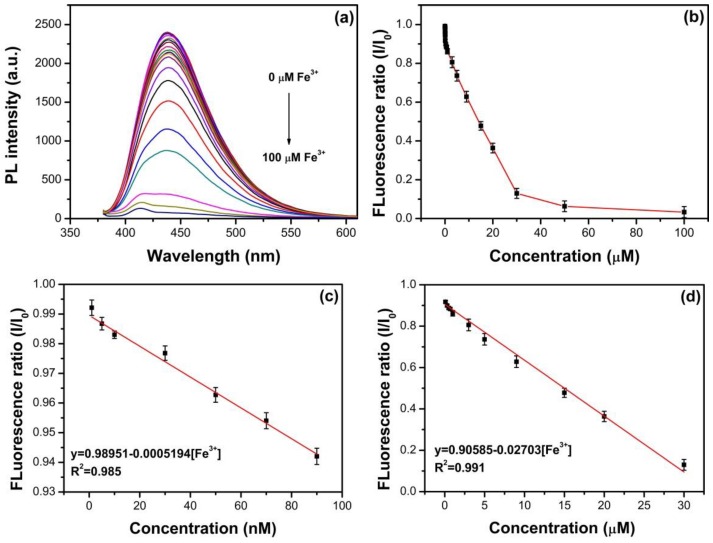
(**a**) Fluorescence response of N, S-GQDs by adding various concentrations of Fe^3+^ ranging from 0 to 100 μM, (**b**) the relationship between fluorescence ratio and Fe^3+^ concentration, (**c**,**d**) the dynamic response range for Fe^3+^ detection in the ranges 1–90 nM and 0.1–30 μM, respectively.

**Figure 12 nanomaterials-09-00738-f012:**
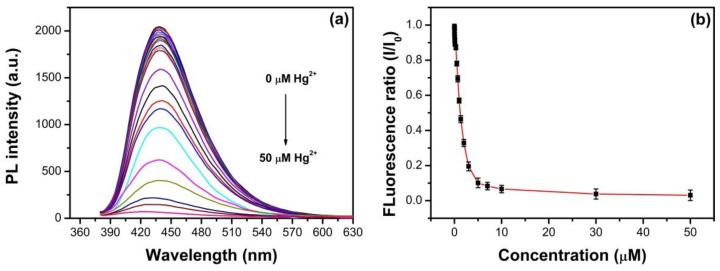
(**a**) Fluorescence response of N, S-GQDs by adding various concentrations of Hg^2+^ ranging from 0 to 50 μM, (**b**) the relationship between fluorescence ratio and Hg^2+^ concentration, (**c**,**d**) the dynamic response range for Hg^2+^ detection in the ranges 1–30 nM and 100–1000 nM, respectively.

**Table 1 nanomaterials-09-00738-t001:** The atomic populations of pure graphene and N, S-GQDs.

Materials		Atomic Populations (Mulliken)				Bond Population		*d* (Å)
		*s*	*p*	Total	Charge (e)			
Pure graphene	C	1.05	2.95	4.00	0	C-C	1.02	1.42
N, S-GQDs	C1	1.13	2.86	3.99	0.01	C1-O	0.64	1.37
	C2	1.07	2.99	4.07	−0.07	C1-S	0.48	1.78
	C3	1.27	2.82	4.09	−0.09	C2-S	0.65	1.71
	N	1.39	3.84	5.23	−0.23	C2-N	0.85	1.39
	O	1.76	4.61	6.37	−0.37	C3-N	0.92	1.37
	S	1.67	3.56	5.24	0.76	C3-O	−0.06	2.40

**Table 2 nanomaterials-09-00738-t002:** Comparison of different fluorescence sensing systems for Fe^3+^ detection. LOD: limit of detection.

Probe	Dynamic Range	LOD	References
Polydopamine dots	10–1000 μM	4.6 μM	[54]
Tyloxapol	0–100 μM	2.2 μM	[55]
Graphene oxide	5–50 μM	0.64 μM	[56]
Dopamine-functionalized GQDs	0.02–20 μM	7.6 nM	[39]
Rhodamine-functionalized GQDs	0–1.0 μM	0.02 μM	[1]
N, S-CQDs	1.5–200 μM	1.02 nM	[57]
GQDs/PS-AER	1–7 μM	0.65 μM	[58]
GQDs	10–200 μM	10 μM	[3]
GQDs	0–60 μM	0.45 μM	[47]
N, S-GQDs	1–90 nM	2.88 nM	This work
	0.1–30 μM	55.49 nM	This work

^a^ GQDs: graphene quantum dots; N, S-CQDs: N and S co-doped carbon quantum dots; GQDs/PS-AER: graphene quantum dots/polystyrenic anion-exchange resin.

**Table 3 nanomaterials-09-00738-t003:** Comparison of different fluorescence sensing systems for Hg^2+^ detection.

Probe	Dynamic Range	LOD	References
Rhodamine derivative	4–15 μM	60.7 nM	[59]
Gold nanoparticle	0.16–1.6 μM	31 nM	[60]
DNA-Ag NCs	2–18 nM	0.25 nM	[61]
CDs	0–3 μM	4.2 nM	[62]
N-CQDs	0–10 μM	1.48 nM	[63]
GQDs	1–50 nM	0.43 nM	[64]
Cysteine-functionalized GQDs	0–10 μM	20 nM	[65]
N-GQDs	0.05–30 μM	1.3 nM	[66]
N, S-GQDs	1–50 nM	0.14 nM	[48]
N, S-GQDs	1–30 nM	0.27 nM	This work
	100–1000 nM	36.85 nM	This work

^b^ DNA-Ag NCs: aptamer DNA silver nanoclusters; CDs: carbon nanodots.

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
