# Peer review of "The Fluorescent Quenching Mechanism of N and S Co-Doped Graphene Quantum Dots with Fe^3+^ and Hg^2+^ Ions and Their Application as a Novel Fluorescent Sensor"

_nanomaterials, 2019, doi:10.3390/nano9050738_

Reviewer 1 Report

            The manuscript titledThe fluorescent quenching mechanism of N and S co-doped graphene quantum dots with Fe3+ and Hg2+ ions and their application as novel fluorescent sensorconcerns the fluorescence intensity of N, S co-doped graphene quantum dots (N, S-GQDs) which can be quenched by Fe3+ and Hg2+. DFT simulation and experimental studies indicate that the fluorescence quenching mechanisms for Fe3+ and Hg2+ detection are mainly attributed to the inner filter effect (IFE) and dynamic quenching process, respectively. The electronegativity difference between C and doped atoms (N, S) in favor to introduce negative charge sites on the surface of N, S-GQDs leads to charge redistribution. Atomic population analysis results show that some charge transfer from Fe3+ and Hg2+ to N, S-GQDs, which relate to the fluorescent quenching of N, S-GQDs. Blue fluorescent N, S-GQDs were synthesized by a facile one-pot hydrothermal treatment. Fluorescent lifetime and UV-vis measurements further validate the fluorescent quenching mechanism which is related to the electron transfer dynamic quenching and IFE quenching.     The manuscript reviews an important analytical topic connected with alternative diagnostic tools.

The paper is well-written, and well-discussed. Due to the fact, manuscript may be ready for publication.

Author Response

Thank you for kindly considering out our manuscript entitled of “The fluorescent quenching mechanism of N and S co-doped graphene quantum dots with Fe3+ and Hg2+ ions and their application as novel fluorescent sensor for drinking water” for publishing in Nanomaterials.

I appreciate very much the positive comments. These comments are very helpful for me to improve the quality of the paper. According to the comments, we have revised the manuscript.

It will be great if you could inform us at your early convenience when it’s received or accepted.

Reviewer 2 Report

Authors have developed a DFT model and show some experimental analysis show the quenching fluorescence due to electron transfer between cations and N, S-GQD. They use the system for detection of Fe and Hg ions and show a reasonable linear results with in the detection limit of 55 nM. They proved that the system they offered can have nanomolar level sensitivity which is widely desired in many filtering systems. They showed these findings using DFT and instrumental analysis such as XPS with cross confirming PL quenching results. Overall, the article is prepared very well, organized, concise and coherent. In addition, It demonstrates new effects of of Phd and Ion concentration on PL intensity. The topic is inserting for a broad range our readers. There is no need for further experiments to be conducted before publication. However, I have several questions before publications.

1)      Can you please discuss the large scale application and device geometry in detail.

2)      Is there any study about the lifetime of the system.

3)      How would you avoid saturation in the system for mass ion detection applications? 

Author Response

(The authors gave the same response as above.)

Reviewer 3 Report

the paper can be published after some editing in English language

Author Response

(The authors gave the same response as above.)
